# Original Research Article

DNA methylation; Whole-genome bisulfite sequencing; machine learning; Differential methylation; Differentially methylated regions; Epigenome.

P.H. and J.H. contributed equally to this work.

**Authors for correspondence:**
S. J. Schultheiss, C. Becker,
E-mail:
sebastian.schultheiss@computomics.com;
claude.becker@biologie.uni-muenchen.de

# MethylScore, a pipeline for accurate and context-aware identification of differentially methylated regions from population-scale plant whole-genome bisulfite sequencing data

Patrick Hüther[1,2] (ORCID), Jörg Hagmann[3], Adam Nunn[4,5], Ioanna Kakoulidou[6], Rahul Pisupati[1], David Langenberger[4], Detlef Weigel[7] (ORCID), Frank Johannes[6,8], Sebastian J. Schultheiss[3,*] (ORCID) and Claude Becker[1,2,*] (ORCID)

[1]Gregor Mendel Institute of Molecular Plant Biology GmbH, Austrian Academy of Sciences, Vienna BioCenter (VBC), 1030 Vienna, Austria; [2]LMU Biocenter, Faculty of Biology, Ludwig-Maximilians-University Munich, 82152 Martinsried, Germany; [3]Computomics GmbH, 72072 Tübingen, Germany; [4]ecSeq Bioinformatics GmbH, 04103 Leipzig, Germany; [5]Department of Computer Science, Leipzig University, 04107 Leipzig, Germany; [6]Department of Plant Sciences, Technical University of Munich, 85354 Freising, Germany; [7]Department of Molecular Biology, Max Planck Institute for Biology, 72076 Tübingen, Germany; [8]Institute for Advanced Study, Technical University of Munich, 85748 Garching, Germany

## Abstract

Whole-genome bisulfite sequencing (WGBS) is the standard method for profiling DNA methylation at single-nucleotide resolution. Different tools have been developed to extract differentially methylated regions (DMRs), often built upon assumptions from mammalian data. Here, we present MethylScore, a pipeline to analyse WGBS data and to account for the substantially more complex and variable nature of plant DNA methylation. MethylScore uses an unsupervised machine learning approach to segment the genome by classification into states of high and low methylation. It processes data from genomic alignments to DMR output and is designed to be usable by novice and expert users alike. We show how MethylScore can identify DMRs from hundreds of samples and how its data-driven approach can stratify associated samples without prior information. We identify DMRs in the *A. thaliana* 1,001 Genomes dataset to unveil known and unknown genotype–epigenotype associations.

## 1. Introduction

Cytosine methylation, which is often used synonymously with DNA methylation, describes the covalent attachment of a methyl group to carbon 5 of cytosine, resulting in 5-methylcytosine (5mC). In plants, as in most eukaryotes, DNA methylation is part of an epigenetic mechanism involved in transposon silencing (Miura et al., 2001), heterochromatin formation (Lippman et al., 2004) and gene regulation (Jaenisch & Bird, 2003). It also plays a role in genome organization (Huff & Zilberman, 2014; Zemach et al., 2013), regulation of development (Finnegan et al., 1996; Papareddy et al., 2021; Ronemus et al., 1996) and imprinting (Gehring et al., 2006; Jullien et al., 2008; Pignatta et al., 2018). DNA methylation can also be a source of phenotypic variation in natural populations (Eichten et al., 2011; Heyn et al., 2013; Schmitz et al., 2013).

Despite recent advances in long-read sequencing, short-read whole-genome bisulfite sequencing (WGBS) is still considered the gold standard for analysis of DNA methylation at single-base resolution. Treatment of genomic DNA with sodium bisulfite causes hydrolytic deamination of unmethylated cytosine (C) into uracil, which upon sequencing library amplification by PCR is converted to thymine (T) (Frommer et al., 1992). The initial step is kinetically inhibited in the case of 5mC, and so methylated cytosines will be interpreted normally during short-read sequencing, whereas unmethylated, converted cytosines will be interpreted

as T by the base caller. Comparison to the known reference genome can subsequently reveal the methylation state of each cytosine.

## 1.1. Features of plant DNA methylation

In many eukaryotic organisms, cytosines can be in a methylated state when they are in a CG sequence context, that is, followed by a guanine (G). In mammalian genomes, the majority of CGs are methylated, while unmethylated cytosines are grouped in so-called CpG islands, which are non-uniformly distributed along the genome and play an important role in transcriptional regulation (Deaton & Bird, 2011). In plant genomes, however, the situation is fundamentally different and can vary substantially from one species to another. DNA methylation in plants occurs in three possible sequence contexts: CG, CHG and CHH (where H is any base but G) (Law & Jacobsen, 2010). Each type of methylation is established, maintained and regulated by a specific molecular machinery (Henderson & Jacobsen, 2007; Law & Jacobsen, 2010; Stroud et al., 2013b). The frequency of methylation along the genome differs by sequence context, for example, in the model plant *Arabidopsis thaliana*, CGs are methylated most frequently relative to the total number of cytosines in that context (~24% of all CGs are 5mCGs), followed by CHG (~7%) and CHH (~1.7%) (Cokus et al., 2008).

It is important to note that—at least in some plant species, including *A. thaliana*—DNA methylation in the different sequence contexts also differs in terms of methylation rate, that is, in the consistency of the methylation status of a given cytosine across different cells. CG cytosines tend to have a binary methylation state, being either always unmethylated or almost always methylated in different cells of a given tissue. In contrast, cytosines in CHG and CHH show more variable methylation status, with mean methylation rates across all methylated CHG and CHH cytosines of ~50 and ~30%, respectively. This has consequences for the analysis of differential methylation between samples (Becker et al., 2011), for example in software which would model the underlying methylation status based on expected distributions (Hansen et al., 2012; Hebestreit et al., 2013; Korthauer et al., 2018).

## 1.2. Whole-genome DNA methylation analysis as a means to understand natural variation and stress response

Since the first characterisations of the whole-genome DNA methylation profile in *A. thaliana* (Cokus et al., 2008; Lister et al., 2008), there has been a growing interest in studying this epigenetic mark at the genomic level to better understand developmental processes (Manning et al., 2006; Pignatta et al., 2018), stress responses (Liu et al., 2018; Wibowo et al., 2016), phenotypic plasticity and natural variation (Kawakatsu et al., 2016; Schmitz et al., 2013). For example, heritable genetic variation alone is not always sufficient to explain the range of phenotypic diversity observed for individuals of the same species, and epigenetic variation is likely to account for at least some of the missing heritability (Manolio et al., 2009). Epigenetic variation, albeit often confounded by genetic variation, has been recognised as a source of natural diversity (Cervera et al., 2002; Riddle & Richards, 2002; Vaughn et al., 2007). As a consequence, some phenotypic traits, including floral symmetry (Cubas et al., 1999), fruit development and morphology (Manning et al., 2006; Ong-Abdullah et al., 2015; Zhong et al., 2013), plant height (Miura et al., 2009 and resistance to pathogens (Liégard et al., 2019)), have been associated with naturally occurring epigenetic alleles (epialleles) that do not appear to be due to linked genetic variation. In addition, studying the conditional dynamics of DNA methyla-

tion at the whole-genome level has highlighted the contribution of epigenetic regulation to stress response and tolerance (Wibowo et al., 2016).

## 1.3. The challenges in determining differential DNA methylation

DNA methylation is highly dynamic, and there is substantial variation among individuals or (sub)populations. It is important to distinguish specific, relevant differences from stochastically occurring methylation differences between biological replicates. On a genome-wide scale, this is challenging without a priori knowledge of stratification, either based on sequence context or by sample grouping. Most experimental studies contrast DNA methylation from different samples to each other, be it mutant background and wild-type, treatment and control, or different natural accessions. Statistical comparisons between samples aim to identify DNA methylation differences at either the single-cytosine (differentially methylated positions [DMPs]) or the region (differentially methylated regions [DMRs]) level. While DMPs provide useful information on the rate at which epigenetic changes occur (Becker et al., 2011; Schmitz et al., 2011; van der Graaf et al., 2015), DMRs are arguably more relevant in a functional biological context because they can affect contiguous stretches of DNA and hence potentially influence the accessibility of regulatory elements. However, the nature of WGBS data imposes several caveats to accurately determining DMPs and DMRs. Some of these reside in the experimental design or the quality of the sequencing library. Insufficient replication, for example, limits the statistical power of differential analyses. Uneven coverage can bias the base calling because of sequencing error rates, while incomplete bisulfite conversion can cause a false estimate of methylated cytosines, as can duplicated reads that arise from library over-amplification.

The statistical analysis of differential methylation often brings along another set of caveats. For example, strategies that are based on defining DMRs as clusters of spatially adjacent DMPs, such as *DSS* (Feng et al., 2014), are subject to a heavy multiple-testing burden resulting from the large number of cytosines in the genome that need to be tested individually. The same applies to window- or sliding-window-based approaches, as implemented, for example, in *methylKit* (Akalin et al., 2012). In contrast, strategies that call DMRs only in predefined regions, for example, in annotated features, risk missing relevant loci in the analysis.

More recently developed tools for DMR calling are centred around pre-selection of genomic regions. *metilene* (Jühling et al., 2016), *dmrseq* (Korthauer et al., 2018) and *HOME* (Srivastava et al., 2019) implement multi-step strategies that first restrict the testable genome space to candidate regions with evidence of methylation, prior to assessing significant differences. Even though *HOME* takes non-CG methylation into account, all three of these tools have been developed with mammalian (mostly human) DNA methylation data in mind and are therefore built on assumptions that do not reflect the distinct genetic control underlying the CG, CHG and CHH methylation contexts in plants. In particular, in mammals, most cytosines are methylated, display strong local correlation between methylation states, are predominantly (or exclusively) methylated in the CG sequence context and exist in largely binary methylation states. This implies that if these assumptions are indiscriminately applied to plant methylomes, many methylated cytosines and regions are likely to be falsely discarded.

Here, we set out to address this gap by developing a tool for the robust identification of DMRs from plant WGBS data that take into account the complexity and variability of plant DNA methylation,

while using an informed, restricted set of candidate regions for statistical testing. Furthermore, we aimed to avoid the necessity for pre-defining sample groups, required by existing tools, to increase the applicability to sample populations with inherent group structure and to prevent experimenter bias. We present MethylScore, a reproducible, stable and self-contained pipeline implemented in Nextflow, that enables biology researchers to conveniently process WGBS data, from read alignments to DMR output. The differential methylation analysis module of MethylScore is built around the two-state hidden-Markov-model-based approach described by Molaro et al. (2011). To identify and segment methylated regions (MRs) in plant genomes, independent of prior information, we extended the original implementation beyond the CG sequence context, allowing the algorithm to train distinct parameter sets for each methylation context (as previously described in Hagmann et al., 2015; see also Supplementary Methods).

MethylScore trains on the different properties of CG, CHG and CHH methylation by estimating parameters of a beta-binomial distribution for each sequence context separately, accounting for both stochastic variance in the coverage distribution (assumed to be beta-distributed) and biological between-sample variance (binomially distributed). In contrast to most mammalian-targeted DMR calling tools, this data-driven approach does not need prior knowledge about the densities and methylation level profiles of the different sequence contexts, as MethylScore learns them from the actual WGBS data. This avoids potentially erroneous or species-specific guesses about the distribution of methylation or arbitrary thresholds. This way, the number of statistical tests is constrained to regions of interest, curtailing the multiple-testing problem. Moreover, its built-in population-scale approach allows identifying DMRs in large datasets, without prior information on sample groups, enabling an unbiased DMR calling. Using publicly available datasets from *A. thaliana* (Cell, 2016; Kawakatsu et al., 2016) and rice (Stroud et al., 2013a), we show that MethylScore is able to segment plant genomes with very different global DNA methylation profiles. In absence of sample information, MethylScore identified group-specific DMRs and was able to detect population signals in datasets with hundreds of samples. Ultimately, we used the DMRs thus identified in the *A. thaliana* 1,001 Genomes and Epigenomes datasets (Cell, 2016; Kawakatsu et al., 2016) to detect known and unknown genotype–epigenotype associations. MethylScore is built as a Nextflow pipeline, allowing ease-of-access and broad usability; the pipeline can be downloaded from https://github.com/Computomics/MethylScore.

## 2. Results

### 2.1. Outline of the modular MethylScore pipeline to call DMRs from plant WGBS data

MethylScore is implemented as a modular pipeline that is built in Nextflow (Di Tommaso et al., 2017) for maximal ease-of-access (Supplementary Figure S1; Supplementary Methods). It uses reference-aligned sequences in *bam* format or tabular single-cytosine information in *bedGraph* format as they are produced by common WGBS primary analysis tools such as *bismark* (Krueger & Andrews, 2011) or *MethylDackel* (https://github.com/dpryan79/MethylDackel). The MethylScore pipeline performs the following general steps: (a) calling of methylated cytosines and determining the per-cytosine methylation rate, (b) identifying contiguous regions with high DNA methylation in each sample, (c) determin-

ing segments as candidates for statistical testing (Supplementary Figure S2), and (d) performing statistical tests to identify DMRs.

### 2.2. Segmentation of plant genomes with different DNA methylation composition

We first wanted to explore how MethylScore would segment plant genomes with very different global DNA methylation configurations. In *A. thaliana*, DNA methylation is unevenly distributed, with most methylated cytosines located in the centromeric and pericentromeric regions (Niederhuth et al., 2016). Overall, 11% of cytosines are methylated, with 31.7% of CG, 16.3% of CHG and 6.2% of CHH cytosines showing methylation rates above zero. In contrast, in the genome of rice (*Oryza sativa*), DNA methylation is more evenly distributed along the chromosomes and more frequent in general, showing 48% CG, 25.3% CHG and 5.7% CHH methylation. When segmenting WGBS data for both species, MethylScore detected 42,478 MRs in *A. thaliana* versus 379,227 in rice, covering 19.5 and 32.5% of the total genome sequence, respectively. Accordingly, the median length of MRs in Arabidopsis (111 bp) was much shorter than in rice (188 bp) (Supplementary Figure S3).

### 2.3. MethylScore identifies DMRs without a priori sample information

We designed MethylScore to be unaware of sample relationships, which means that it does not require any a priori information about replicate groups and that the analysis will not be biased by potentially inaccurate assumptions about sample similarities. Instead, for each candidate DMR, MethylScore groups all samples using an iterative *k*-means approach (see Section 5), followed by a beta-binomial-based test to statistically assess whether the group means are significantly different from each other. As a result, grouping may not necessarily reflect the preconceived biological design of the experiment for each candidate DMR, yet local grouping of DMRs has the advantage that it allows to identify patterns in the underlying sample structure that might be overlooked when defining groups beforehand. Averaged over all candidate DMRs, methylation differences should be apparent between, for example, treatment groups or developmental stages *if* the underlying hypothesis for these groups to differ in DNA methylation was correct. To test if MethylScore accurately differentiates between-group and within-group differences in this unsupervised approach, we applied it to *A. thaliana* datasets that had previously been described as having substantial levels of between-group methylation divergence.

We first analysed a relatively simple WGBS dataset of Columbia-0 (Col-0) wild-type and two loss-of-function mutants of *EFFECTOR OF TRANSCRIPTION 1* (*et1-1*) and *EFFECTOR OF TRANSCRIPTION 2* (*et2-3*), respectively, as well as of an *et1-1 et2-3* double mutant, with three biological replicates per genotype. These mutants had previously been shown to have local DNA methylation changes compared to wild-type but also compared to each other (Tedeschi et al., 2019). When running MethylScore on all samples without group replicate information, methylation differences in DMRs accurately reflected the genotypic relationship, indicating that the grouping of samples over all DMRs was driven by methylation differences between genotypic groups (Figure 1). In accordance with the original publication, grouping was most conspicuous in CG and CHG contexts (Figure 1).

Next, we asked whether MethylScore would accurately classify methylation pathway mutants with more pronounced DNA methylation differences to wild-type. Some of these mutants have

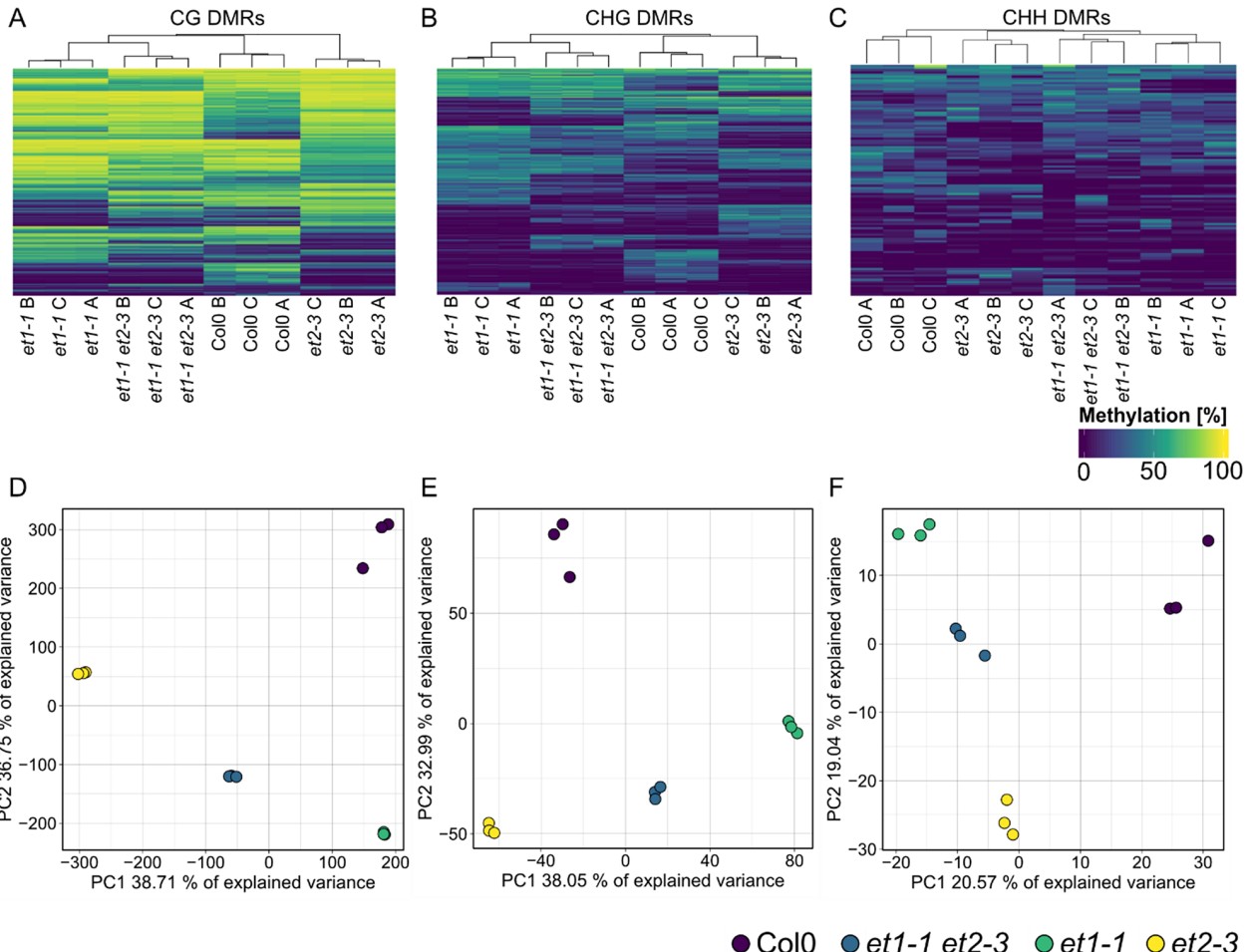

**Fig. 1.** Methylation within DMRs assigns samples into groups according to genotype. Heatmaps and principal component analyses of mean methylation rates in 9,487 CG- (a,d), 1,282 CHG- (b,e) and 741 CHH- (c,f) context-specific DMRs identified by MethylScore from WGBS data from flowers of *A. thaliana* Col-0 wild-type, respective *et1-1* and *et2-3* single mutants, and *et1-1 et2-3* double mutants. Original data from Tedeschi et al. (2019) (ENA accession PRJEB12413).

substantial global loss of cytosine methylation; we therefore wanted to see whether the training of the HMM, which takes into consideration the methylation rate distributions of each sample, might be affected by this. We used a dataset comprising loss-of-functions mutants of the CHG-specific DNA methyltransferase CMT3, the chromatin remodeler DECREASE IN DNA METHYLATION 1 (DDM1), two regulators of these genes, TESMIN/TSO1-LIKE CXC DOMAIN-CONTAINING PROTEIN 5 (TCX5) and TCX6, and of combinations thereof (Ning et al., 2020). Similar to the *et* mutant analysis (Figure 1), MethylScore accurately clustered the samples according to genotype based on DNA methylation rates within DMRs (Figure 2). Context-specific DMRs also resolved CG- and CHG-specific loss of methylation in *ddm1* and *cmt3*, respectively (Figure 2), indicating that deviations from the standard methylation rate distributions did not affect MethylScore analysis. We compared DMR calling in the *cmt3* background versus wild-type between MethylScore and two context-unaware DMR callers, *DSS* (Feng et al., 2014) and *metilene* (Jühling et al., 2016) (Supplementary Figure S4). While many of the DMRs detected by the respective tools overlapped, each tool identified a 'private' set of DMRs that were specific for that tool. *DSS*, which uses DMP clustering to identify DMRs, detected hardly any CHH-DMRs, while metilene detected fewer DMRs than MethylScore in all contexts. MethylScore DMRs were on average shorter than those of the other

tools, which, in combination with the finding that many regions overlapped, might indicate sharper region boundaries (Supplementary Figure S4).

### 2.4. Clustering of MRs across regenerated A. thaliana populations underlines partial maintenance of organ-specific methylation profiles in CG

Mutants known to be affected in DNA methylation arguably are expected to present relatively simple cases for genome-wide differential analyses. To test more complex situations, we applied MethylScore's population-scale approach to more complex datasets with less predictable group structure. We re-analysed a published dataset of *A. thaliana* regenerants derived from different tissues of origin that had been obtained by induction of somatic embryogenesis (Wibowo et al., 2018). Instead of employing a pairwise testing strategy between condition groups as it had been pursued in the original study, we used MethylScore to pre-select candidate regions based on MR frequency changes across the sample population and hypothesised that such a strategy should naturally find clusters of regenerants with a similar epigenetic setup, thus allowing to differentiate between organ-derived methylation signatures. The data contained three potentially interacting factors that could contribute to epigenetic variation: regeneration via somatic embryogenesis

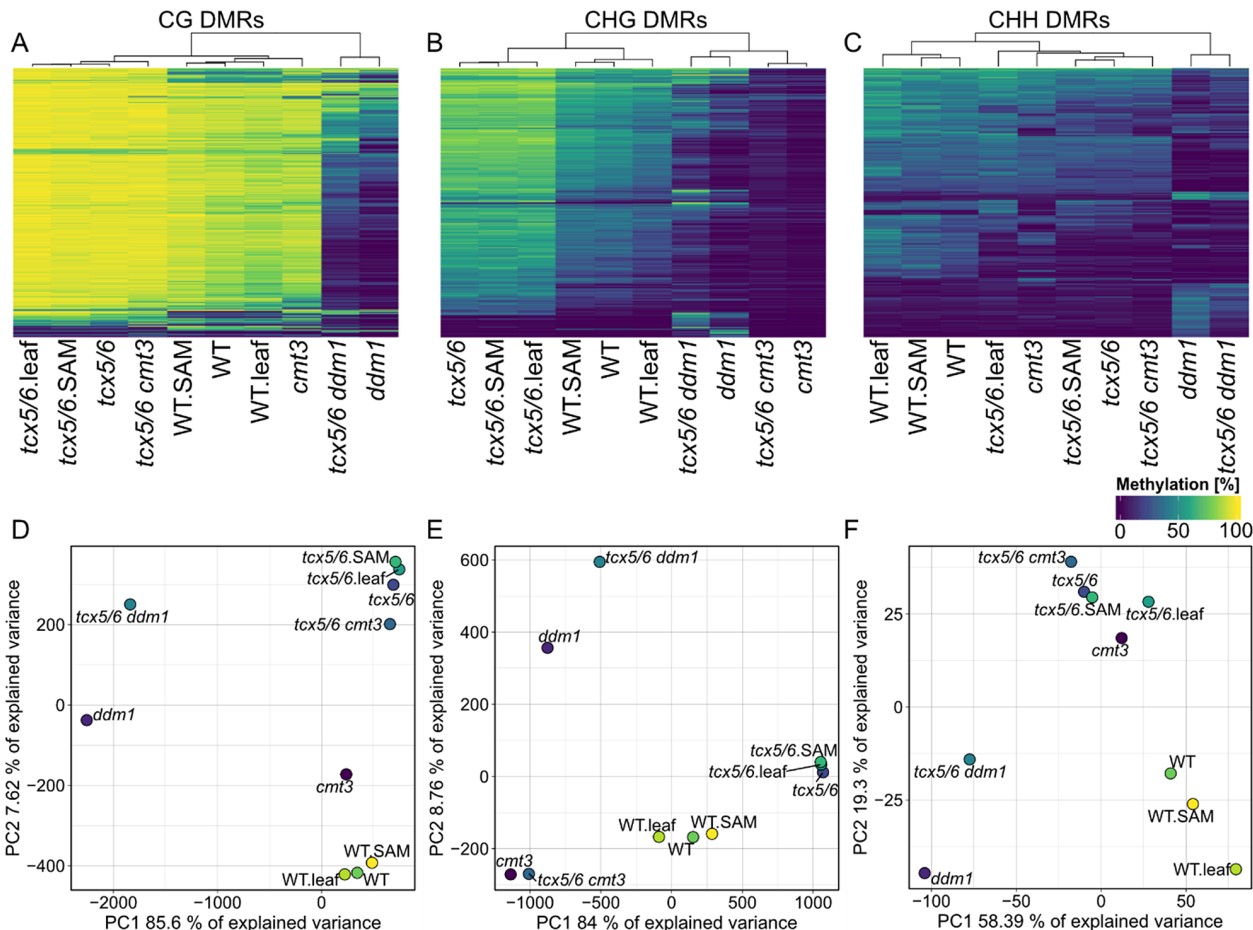

**Fig. 2.** Unsupervised DMR calling from WGBS data of DNA methylation pathway mutants. Heatmaps and PCAs of mean methylation rates in 59,153 CG- (a,d), 63,385 CHG- (b,e) and 440 CHH- (c,f) context-specific DMRs identified by MethylScore from WGBS sequencing data of DNA methylation pathway mutants. The dataset included *ddm1* and *cmt3* single mutants, *tcx5/6* double mutants, as well as *tcx5/6 ddm1* and *tcx5/6 cmt3* triple mutants. DNA had been sampled from leaves and shoot apical meristem (SAM). Original data from Ning et al. (2020); GEO accession GSE137754.

versus sexual reproduction; tissue of origin of the somatic embryos (RO, root origin; LO, leaf origin); and tissue of DNA sampling (leaf or root) in the progeny of the regenerants.

Hierarchical clustering of methylation rates in 1,282 CHG- and 741 CHH-DMRs revealed tissue type during DNA sampling as the major determinant of methylation within these regions (Figure 3a,b). Tissue of origin was subordinated to sampling tissue type. Principal component analysis (PCA) of methylation rates within the same regions also showed clear ordination based on sampling tissue type along PC1 (Figure 3b), for both CHH and CHG contexts. In contrast, among all 9,487 DMRs identified in CG context, tissue of origin appeared to play a role alongside sampling tissue type. Our analysis also recapitulated the main finding of the original publication, namely that DNA methylation in leaves of RO progenitors was more similar to that in root tissue. Our results generally supported the original conclusion of partial maintenance of epigenetic marks retained from the tissue of origin across generations of sexually reproduced offspring, yet suggested CG methylation to be the main driver of this observation (Figure 3a).

To validate whether MethylScore could accurately resolve complex data structures in larger plant genomes with much denser DNA methylation profiles, we also re-analysed a regeneration-related dataset from rice (Stroud et al., 2013a). Similar to the *A. thaliana* dataset, MethylScore identified DMRs that separated

regenerated from non-regenerated samples (Supplementary Figure S5). Interestingly, however, clustering of samples based on methylation rates within DMRs changed considerably depending on the sequence context. For example, callus samples were similar to control plants and regenerants in CG-DMRs but stood out as hypermethylated in the CHH context (Supplementary Figure S5).

### 2.5. DMR calling on a population scale can identify unknown underlying sample structures

MethylScore does not require information on sample grouping but instead clusters samples for each DMR based on methylation rates, so we expected it to be able to call DMRs even in very large population-scale datasets. We also wanted to explore how combined epigenetic and genetic variation would affect the algorithm. Natural genetic and epigenetic variation in *A. thaliana* has been cataloged in the 1,001 Genomes and Epigenomes Project (www.1001genomes.org) (Cell, 2016; Kawakatsu et al., 2016). We analysed a subset of 645 *A. thaliana* accessions that had been sequenced by WGBS at the SALK Institute, San Diego, CA. Applying the population-scale approach of MethylScore, we identified 60,797, 16,627 and 8,406 DMRs in CG, CHG and CHH, respectively. PCA on the accession-specific methylation rates in these regions revealed geographic clustering that separated a cluster of

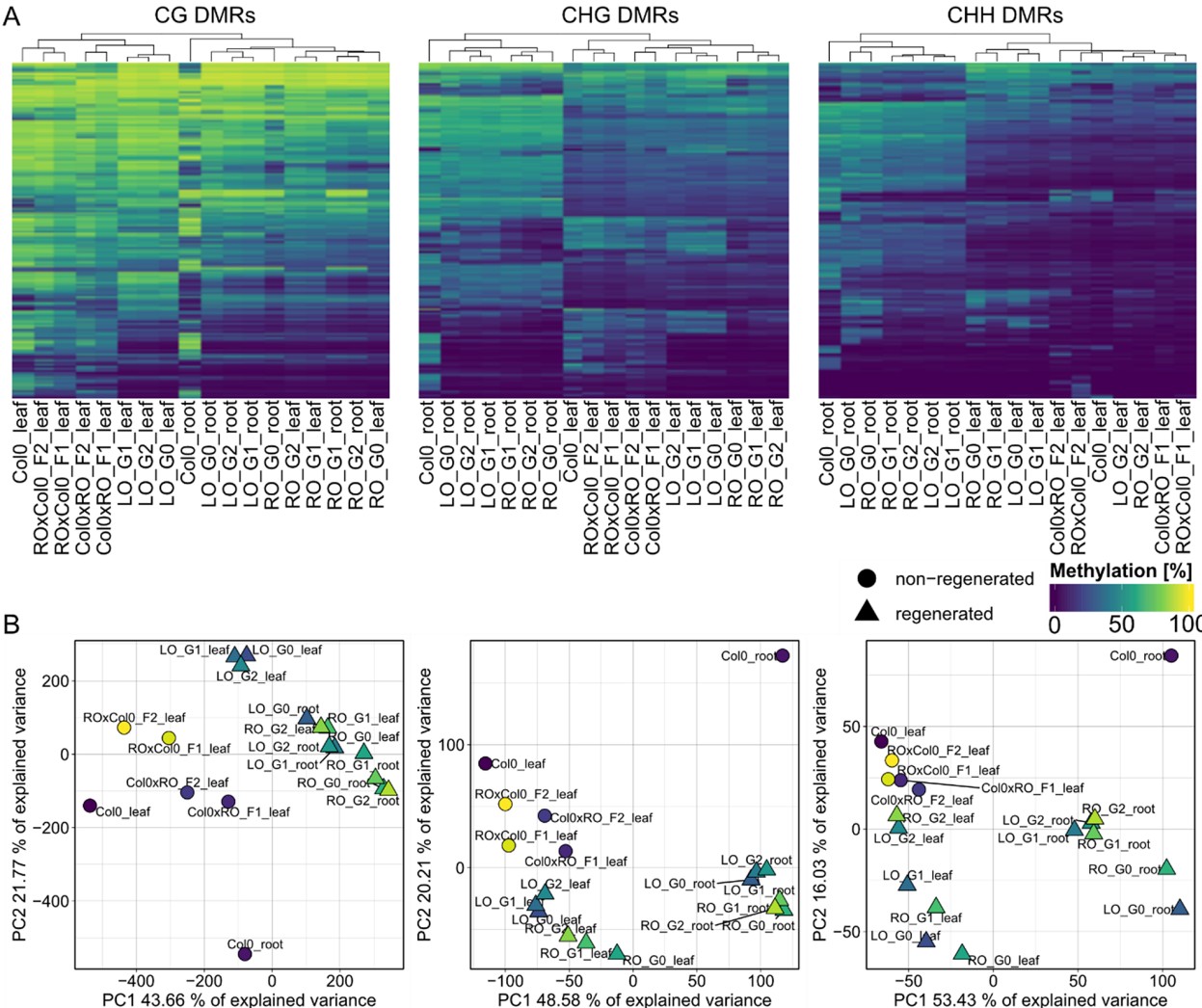

**Fig. 3.** MethylScore population clustering partially reflects epigenetic origin of regenerated plant lineages. (a) Heatmaps show methylation rate averages in regions identified as differentially methylated in CG, CHG or CHH methylation contexts (from left to right). The dataset includes leaf and root tissue from Col-0 control plants as well as from generation 1 (G1) and generation 2 (G2) progeny of somatic regenerants from root origin (RO) and leaf origin (LO) somatic embryos, and leaf tissue from F1 and F2 backcrosses of RO and LO regenerants to Col-0. (b) PCAs for each methylation context using the same data are shown in (a). Original data from Wibowo et al. (2018), ENA accession PRJEB26932.

Central Asian accessions from the rest (Figure 4). Probably due to the large sample number, the total variance explained by any principal component remained relatively low (Figure 4). This grouping occurred for all three sequence contexts and had not been described in the original publication, indicating that MethylScore's approach can detect data structures that remain hidden with conventional DMR calling tools. Moreover, CG, but not CHG or CHH DMRs, resolved a latitudinal transect for European accessions (Figure 4). We wanted to explore the source of the signal further and used the geographic coordinates of collection sites for each of the *A. thaliana* accessions to retrieve bioclimatic variables from the WorldClimate dataset (www.worldclim.org) (Fick & Hijmans, 2017). While the Central Asian cluster showed strong correlation with low annual average temperature (bio1), strongest correlation was observed for the lowest temperature in the coldest month of the year (bio6). While local climate is strongly confounded by geographic location and thus population structure, this observation is in line with previous reports that showed interdependence between DNA methylation and ambient temperature (Dubin et al., 2015) as well as seasonality (Shen et al., 2014).

To further explore MethylScore's capability to resolve hidden sample/population structures, we applied it to WGBS data of 169 epigenetic recombinant inbred lines (epiRILs) (Zhang et al., 2021). These lines are derived from a cross of Col-0 wild-type and a loss-of-function mutant of DDM1 and propagated by single-seed-descent. Genetically, these lines can be considered near-isogenic; previous studies showed that these epiRILs carry marked differences in DNA methylation in mosaic-like patterns of DNA methylation, some of which are stably inherited through generations of single-seed descent (Colomé-Tatché et al., 2012; Johannes et al., 2009). As every epiRIL has been propagated independently and should therefore be independent from every other line, we did not expect any sample relationships. However, when analysing the DMRs returned by MethylScore's population-scale analysis in a PCA, the epiRILs formed several clusters (Supplementary Figure S6). In all three sequence contexts, most lines were contained in one large cluster, with the exception of a few lines that clustered separately (Supplementary Figure S6). This clustering may reflect the fact that lines were selected for WGBS based on phenotypic information (i.e., selected epigenotyping) or yet unknown

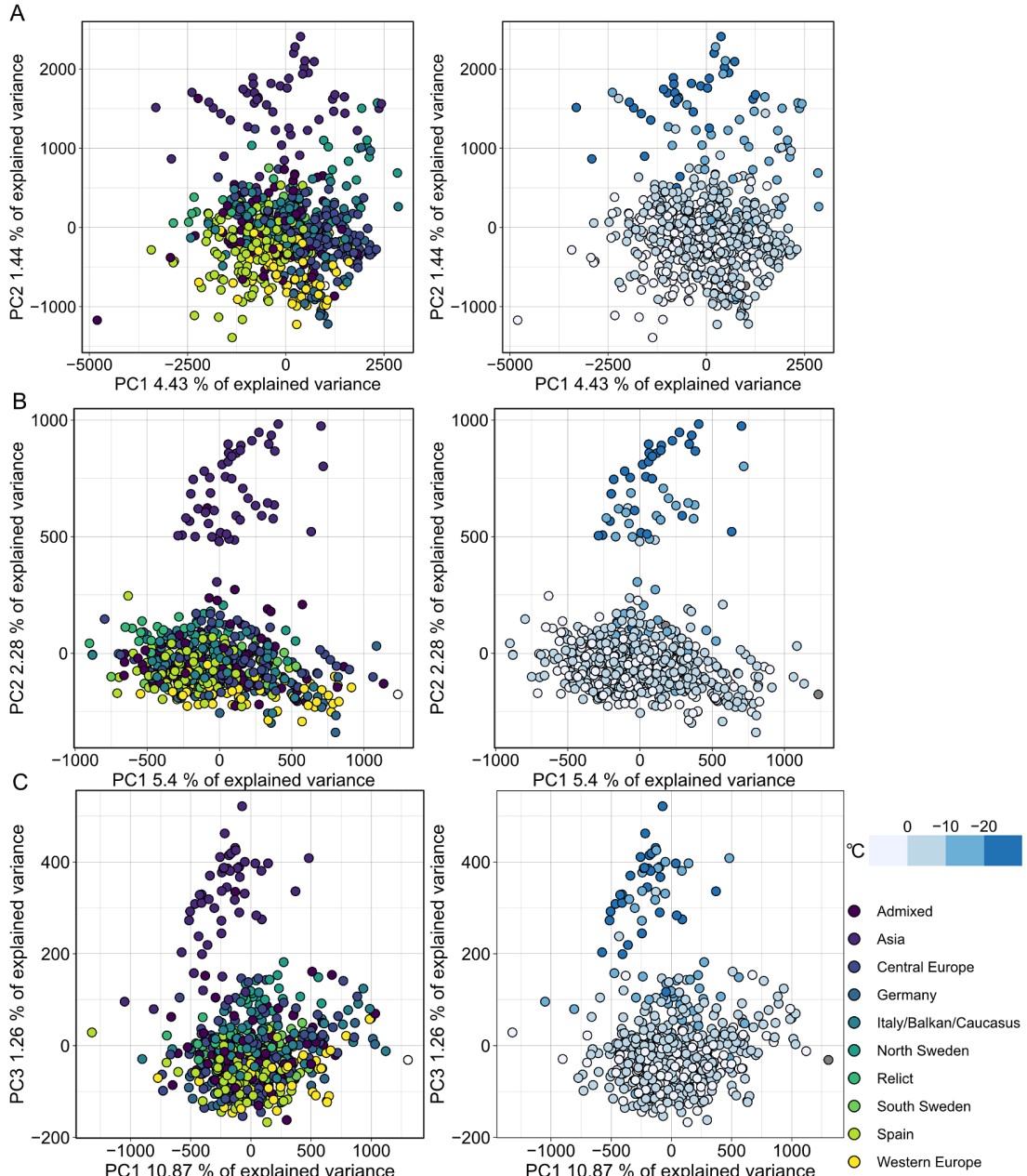

**Fig. 4.** Population structure analysis of natural *A. thaliana* accessions based on DMRs identified by the population-scale clustering approach of MethylScore. PCA shows group formation in CG (a), CHG (b) and CHH (c) methylation contexts. Colours indicate admixture groups (left column) and seasonality with regard to the lowest temperature in the coldest month (right column). Data were retrieved via geographic coordinates of collection sites for each accession from the worldclim.org bio6 dataset (Fick & Hijmans, 2017). Original WGBS published in Kawakatsu et al. (2016).

sources of sample stratification, including genetic structure due to shared TE insertion profiles (Quadrana et al., 2019) in a subset of lines.

### 2.6. MethylScore DMRs reveal recurrent trans-acting association signals in A. thaliana natural accessions

MethylScore's *k*-means clustering leads to grouping of samples into only a small number of groups (mostly two or three); hence, within-group variance becomes relatively large when sample numbers are high, as was the case for the 1,001 Genomes data. As a result, DMRs with noticeable between-group differences become rare, explaining the relatively low number of DMRs we observed in the 1,001 Genomes data, in relation to the size of the dataset. To increase the statistical power to detect genotype—epigenotype associations

in a genome-wide association (GWA) mapping approach that uses methylation rates in each DMR as the phenotype vector, we changed from a population scale to a pairwise DMR calling, comparing each of the 645 accessions individually to the Col-0 reference. On average, each pairwise comparison identified 1,708, 778 and 3,625 DMRs in CG, CHG and CHH contexts, respectively. Many DMRs from different pairwise comparisons partially or fully overlapped, so we created sets of unions of overlapping DMRs for each context, resulting in 18,044 CG-, 9,674 CHG- and 25,350 CHH-DMRs.

We queried methylation rates per accession and conducted GWAS analyses, using region-level methylation averages as the phenotype vector. On the genotype level, we considered all geno-typed SNPs among the 645 accessions that had a minor allele

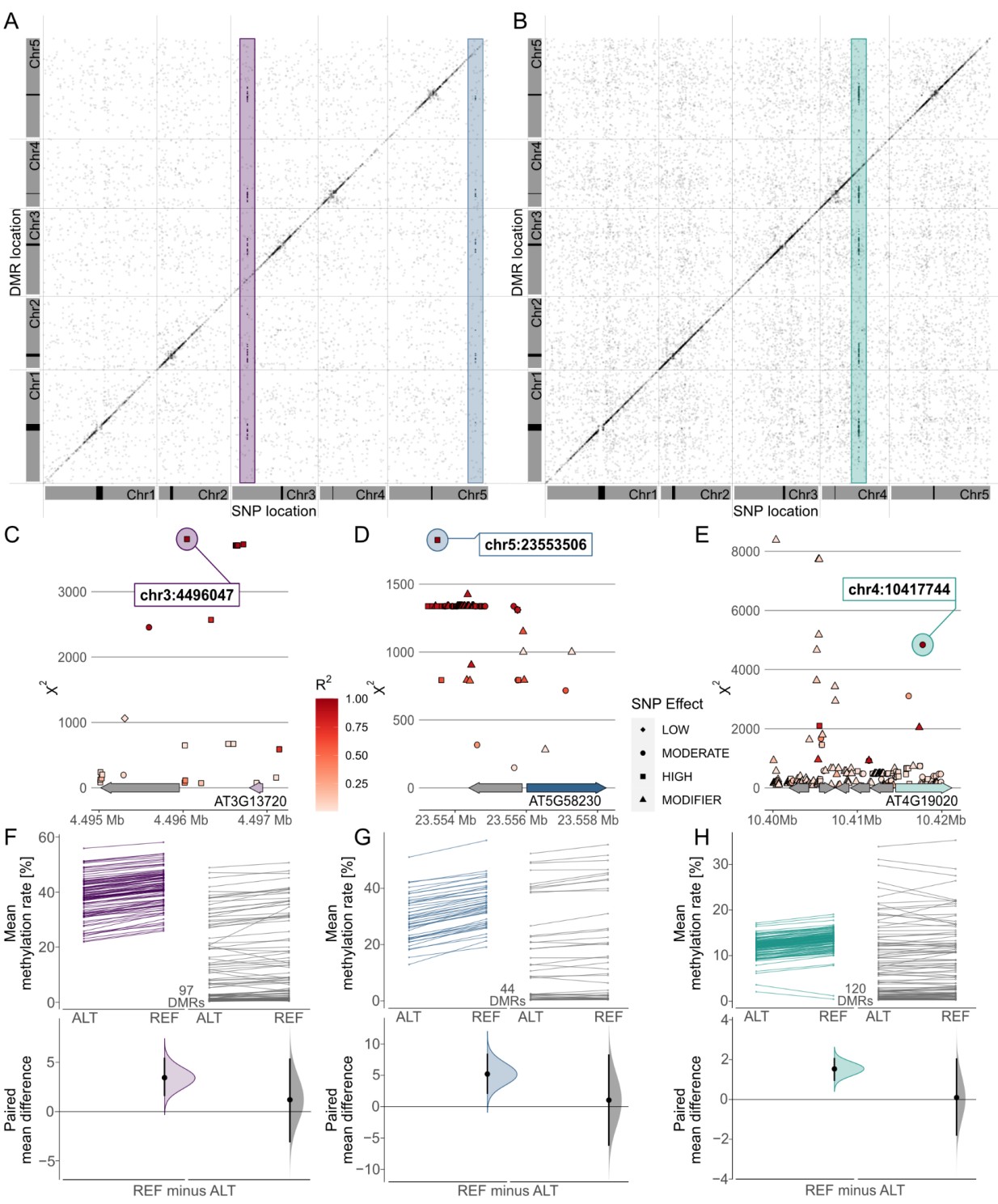

**Fig. 5.** Genome wide association (GWA) signals recurrently emerge from differential methylation found across the *A. thaliana* 1,001 Methylomes panel (Kawakatsu et al., 2016). GWA analyses on region-level methylation rate averages reveal recurrent signals in CHG (a) and CHH (b) methylation contexts. For each DMR, only top ranked SNPs that pass the Bonferroni corrected significance threshold at $\alpha = 0.05$ are included, based on the number of SNP markers available across all 646 *A. thaliana* accessions used in the study (1,813,837 SNPs with minor allele frequency >5%, $p < 2.8 \times 10^{-8}$). (c–e) Genomic loci of recurrent trans-acting SNPs highlighted in (a) and (b). (f–h) Effect sizes of SNPs highlighted in (a) and (b) on methylation rates in regions underlying the SNP association are shown as slopegraphs and bootstrap estimates for carriers of the alternative (ALT) and reference (REF) alleles, respectively. In each plot, an equally sized set of randomly selected DMRs (in gray) is included for comparison.

frequency >5% (1,813,837 SNPs). Using this strategy, we identified 16,083 regions in CG, 8,102 in CHG and 19,388 in CHH context that yielded at least one SNP association with significance levels that passed the Bonferroni-corrected significance threshold at $\alpha = 0.05$ (threshold = $2.8 \times 10^{-8}$).

Combining these results in a meta-analysis revealed that many of these associations marked region-proximal SNP locations in *cis* for CG (Supplementary Figure S7), CHG and CHH (Figure 5) methylation contexts. Intriguingly, a few also tagged several recurring association signals in *trans* (Figure 5). For CHG DMRs, we

found two recurrent associations with markers on chromosomes 3 and 5, respectively (Figure 5a,d,e). Chr3:4,496,047 was found to be associated with mean methylation rates in 97 DMRs with consistently decreased methylation in accessions carrying the alternative allele (Figure 5f,g). The SNP was located in close proximity to the *miR823A* locus, which encodes the primary miRNA (primiRNA) of microRNA (miRNA) 823A, known to target *CMT3*. *CMT3* is the main methyltransferase depositing CHG methylation (Lindroth et al., 2001) and via its chromodomain provides a direct link between DNA methylation and H3K9 dimethylation at constitutive heterochromatin in a concerted feedback loop with KRYPTONITE (KYP) and *SUPPRESSOR OF VARIATION 3-9 HOMOLOGUE PROTEIN 5/6* (*SUVH5/6*) (Du et al., 2012, 2014; Ebbs & Bender, 2006; Jackson et al., 2002).

The second CHG-DMR-associated locus, Chr5:23,553,506, showed association in 44 DMRs, similarly with relatively lower methylation levels in accessions carrying the alternative allele. The SNP resolved a genomic region upstream of *MULTICOPY SUPPRESSOR OF IRA1* (*MSI1*) and appeared to be in linkage with several markers in that region (Figure 5d). MSI1 acts in the evolutionarily conserved retinoblastoma pathway and is implicated in genomic imprinting of the *FLOWERING WAGENINGEN* (*FWA*) and *FERTILISATION INDEPENDENT SEED 2* (*FIS2*) in *A. thaliana* via direct interaction with *RETINOBLASTOMA RELATED 1* (*RBR1*) protein (Jullien et al., 2008). Interestingly, both RBR1 and MSI1 have also been identified as integral components of the *Arabidopsis* DREAM complex (Ning et al., 2020).

In the CHH context, our finding of a recurrent association at Chr5:10417744 (Figure 5b,e) confirms a recent study that identified natural alleles of *CHROMOMETHYLASE2* (*CMT2*) as determinants of CHH methylation (Sasaki et al., 2019). Similarly, associations of transposon methylation in CHG context and the *miR823* and *MSI1* loci were recently reported (Sasaki et al., 2022). The association signal in CHH emerged from 120 DMRs, showing a general trend of lower methylation in accessions carrying the alternative allele, with the exception of two regions of overall low CHH methylation (Figure 5h). In summary, this shows that MethylScore provides context-specific DMRs from large WGBS datasets that can be used to determine specific genome–epigenotype associations.

## 3. Discussion

Genome-wide studies of DNA methylation typically aim to reveal differences in DNA methylation between samples or groups of samples. This includes identification of phenotypically relevant epialleles, finding epigenetically controlled regulatory loci or assessing naturally occurring epigenetic variation. Here, we have presented MethylScore, a pipeline for the identification and characterisation of differentially methylated loci specifically from plant WGBS data. We built MethylScore in such a way as to make it accessible to a broad community of plant researchers, requiring minimal computational background; provided with alignment files or methylation metrics files and a simple sample information table, MethylScore can be run with a single command line.

Currently available DMR callers are mostly built on assumptions based on mammalian DNA methylation, including the sequence context, genomic distribution and frequency of methylated cytosines and the average methylation rate. Most of these characteristics differ from species to species, and in particular DNA methylation in plants typically occurs in three rather than in only one sequence context (as is the case in mammalian genomes). MethylScore generalises DMR calling to be applicable to many different species by accounting for all three sequence contexts and

by its unsupervised calling of MRs based on a model that is trained on the species' characteristic methylation densities and rates of each sequence context.

Moreover, most DMR callers typically follow one of three general strategies: DMP clustering, genome tiling or pre-defining regions of interest, each with their respective caveats. DMP clustering first tests for DMPs and merges spatially adjacent loci to define DMRs. Examples of such tools are *DSS* (Feng et al., 2014), *Bisulfighter* (Saito et al., 2014), *MOABS* (Sun et al., 2014), *BSmooth* (Hansen et al., 2012) and *methylpy* (Schultz et al., 2015; Ziller et al., 2013). DMP analysis typically involves statistical tests on each cytosine in the genome, racking up a heavy multiple-testing burden. If no appropriate region-level testing is applied, this multiple-testing problem is carried over to the DMRs, resulting in inappropriate false discovery rate (FDR) control caused by significance thresholding at single loci (Korthauer et al., 2018). In addition, many such approaches classify DMRs based on mere presence of DMPs without considering the directionality of methylation change. Multiple-testing limitations also apply to tiling approaches in which testable segments correspond to windows or sliding windows along the genome, implemented, for example, in *methylKit* (Akalin et al., 2012). The third category of DMR callers uses predefined regions, selected based on existing knowledge, for example, genome annotation features. These can be gene bodies, promoter regions, transposons or predetermined CpG islands. While this reduces the multiple-testing problem, it ignores potentially relevant loci that are not included in the predefined set, yet makes tools such as *BiSeq* (Hebestreit et al., 2013) suited for targeted sequencing approaches such as reduced-representation bisulfite sequencing (Meissner et al., 2005).

All of the above methods have additional downsides when applied to *plant* WGBS data. Methylation occurring in all sequence contexts in plants aggravates the DMP multiple-testing problem. Some approaches, including *BSmooth* (Hansen et al., 2012), *BiSeq* (Hebestreit et al., 2013) and *dmrseq* (Korthauer et al., 2018), moreover assume local correlation between spatially adjacent methylated cytosines to compute smoothed methylation estimates. Despite the presence of CHH islands that have been reported for some species (Gent et al., 2013; Zemach et al., 2010), this assumption primarily holds true for studies in mammalian genomes with high density of methylated CpG regions. It is, however, less applicable in plants and potentially affects the performance of these methods in determining genomic boundaries of MRs. Similarly, window- or tile-based approaches are best applicable when methylation is evenly and densely distributed. However, in *A. thaliana* and other plant genomes, the overwhelming share of methylated cytosines is concentrated in only a small fraction of the genome. Window-based approaches therefore cause a substantial penalty in statistical power when controlling for FDR, because many of the statistical tests lack biological relevance in absence of methylation in the windows that are tested.

### 3.1. MethylScore does not require a priori information on DNA methylation parameters or dataset structures

In an unsupervised training step, MethylScore identifies the methylation rate distribution per sequence context in the actual data, followed by classifying the genome into states of high and low methylation based on these training parameters. Alternatively, users can decide to train MethylScore on a reference sample and apply the training parameters on all other samples in their dataset. Only those regions are tested for differential methylation in which high-methylation states differ across samples, and so our pipeline

avoids testing regions with no or very little DNA methylation variation across samples, which drastically reduces the number of necessary statistical tests compared to other methods. We note, however, that in species that are more densely methylated than *A. thaliana* or *O. sativa*, the genomic space contained in MRs will be larger, potentially necessitating more statistical tests.

As shown in the test cases above, MethylScore can handle small datasets with only few samples to very large datasets with hundreds of WGBS runs. Unless users decide to provide information on replicate groups, the pipeline applies a population-scale analysis, clustering samples into groups with similar methylation rates for each testable genomic segment. This has two effects: first, MethylScore is thus able to reveal hidden sample structures that researchers might not be aware of at the start of the study. Second, substantial within-group variance, for example, within a group of replicates of the same treatment condition, will become apparent in the DMR output and will not be masked by a forced a priori grouping of replicates. Nevertheless, MethylScore optionally enables imposing such sample grouping simply by using shared sample identifiers when running the pipeline; this might be desirable in some cases, for example, in mutant-versus-wild-type comparisons.

### 3.2. A modular pipeline that can be integrated with other WGBS analysis pipelines

MethylScore is designed as a post-alignment, secondary analysis workflow starting from either alignments in *bam* format or pre-existing single-cytosine metrics in *bedGraph* format, in contrast to existing end-to-end solutions such as *wg-blimp* (Wöste et al., 2020) or *PiGx bsseq* (Wurmus et al., 2018). However, its modular design also allows for tight integration with primary analysis pipelines such as *nf-core methylseq* (Ewels et al., 2020), *snakePipes WGBS* (Bhardwaj et al., 2019) or *EpiDiverse Toolkit* (Nunn et al., 2021). These pipelines take raw BS-sequencing reads as an input and perform extensive quality control, along with trimming and read mapping, which is highly advised to rule out or mitigate potential biases arising from experimental factors such as library preparation protocols, adapter content and sequence duplication levels, before attempting to identify DMRs.

### 3.3. Limitations and shortcomings of the pipeline

MethylScore detects MRs in a largely unsupervised manner that we expect to generalise to most species. While we show that the default parameter set for the subsequent selection of candidate DMRs is suitable on data derived from rather sparsely methylated *A. thaliana* and more densely methylated *O. sativa*, the filtering of DMRs before testing them depends on a number of parameters (see Supplementary Methods) that can be adjusted depending on the species and/or research question at hand. Of these, however, we expect only two to have a larger impact on the number of called DMRs: the change of MR frequency along the genome, and the mean methylation difference between sample clusters (Supplementary Figure S9). The former controls a minimum epiallele frequency requirement across the sample population, whereas the latter sets a lower boundary for methylation differences. Both thus act as DMR filters and should be set in accordance with the user's interest: DMRs with a low epiallele frequency and/or DMRs with little methylation difference between sample groups can be discarded early on without even testing them. This may introduce a user-dependent bias, but at the same time likely helps to retain more biologically meaningful DMRs.

## 4. Outlook

Despite recent advances in detecting chemically modified bases in long-read sequencing data, WGBS remains the current standard for the analysis of whole-genome DNA methylation profiles. This applies even more to plants, for which the complexity and diversity of base modifications pose major hurdles to long-read-based technologies. Even if these technical issues will be overcome in future, statistical analysis of differential methylation to determine relevant epiallelic loci will remain a key challenge in this type of studies.

## 5. Materials and methods

Please see Supplemental Material for a detailed description of the pipeline and methods.

## Acknowledgements

We would like to thank Niklas Schandry, Eva Knoch and Daniela Ramos Cruz for critical reading of the manuscript and helpful suggestions. The computational results presented were obtained using the CLIP cluster at Vienna BioCenter (VBC) and the BioHPC Genomics cluster housed at the Leibniz Supercomputing Centre (LRZ).

**Financial support.** This research was supported by the Austrian Academy of Sciences (P.H., R.P., C.B.); the European Union's Horizon 2020 research and innovation program via the European Research Council (ERC) Grant agreement No. 716823 'FEAR-SAP' (P.H., C.B.), DFG project ERA-CAPS AUREATE (D.W.) and via the Marie Sklodowska-Curie ETN 'EpiDiverse' (A.N., C.B.), grant agreement no. 764965 'Epidiverse'. F.J. acknowledges support from the Technical University of Munich-Institute for Advanced Study funded by the German Excellent Initiative and the European Seventh Framework Program under grant agreement no. 291763. I.K. and F.J. were supported by the SFB Sonderforschungsbereich924 of the Deutsche Forschungsgemeinschaft (DFG).

**Conflict of interest.** The authors declare the following competing interests: J.H. is currently an employee of Computomics GmbH. S.J.S. is currently the CEO of and holds shares in Computomics GmbH. D.W. hold shares in Computomics GmbH. A.N. is currently an employee of ecSEQ Bioinformatics GmbH. D.L. is currently the CEO of and holds shares in ecSEQ Bioinformatics GmbH.

**Authorship contributions.** P.H., J.H., D.W., F.J. and C.B. conceived the study and developed the pipeline; D.L. and C.B. supervised the work. J.H. wrote the MethylScore code; P.H. built the Nextflow pipeline. I.K. generated the epiRIL data. P.H. and A.N. analysed the data. R.P. prepared methylation calls on the *A. thaliana* 1,001 Genomes dataset. P.H. and C.B. wrote the manuscript.

**Data availability statement.** The MethylScore pipeline is available on Github: https://github.com/Computomics/MethylScore. The pipeline that was used to conduct GWA studies is available on Gitlab: https://gitlab.lrz.de/beckerlab/gwas-nf.

**Supplementary Materials.** To view supplementary material for this article, please visit http://doi.org/10.1017/qpb.2022.14.

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
