## [Reviewer Report]

Dear Editor,

We would like to submit our manuscript entitled “MethylScore, a pipeline for accurate and context-aware identification of differentially methylated regions from population-scale plant WGBS data” (bioRxiv DOI: 10.1101/2022.01.06.475031) for publication in Quantitative Plant Biology.

The last decade has seen a surge of studies that use whole-genome bisulfite sequencing (WGBS) data to study genome-wide profiles of DNA methylation in plants. Most of these studies involve sample or group comparisons, which have the aim to identify biologically relevant differential methylation between samples or groups of samples. Loci classified as differentially methylated regions (DMRs) constitute for example epialleles that underlie phenotypic variation (e.g., Ong-Abdullah et al., Nature 2015) or epigenetic regulatory regions relevant in stress response (e.g., Wibowo et al, eLife 2016). However, confidently detecting such DMRs from WGBS data remains challenging, particularly in WGBS data from plants. First, DNA methylation in plants is highly complex and can occur in three different sequence contexts, each with its own characteristics. Second, DNA methylation levels and frequencies can differ substantially between different plant species. Finally, current DMR callers were developed based on assumptions that apply to mammalian DNA methylation and that often do not apply to the characteristics of DNA methylation in plants.

Here, we present MethylScore, a pipeline designed to call DMRs from WGBS alignment files specifically from plant WGBS data. Our tool uses an unsupervised Hidden-Markov-Model-based approach to segment plant genomes into high and low methylation states, specific for each sequence context. To this end, it learns context-specific DNA methylation distributions from the actual WGBS data. MethylScore does not require sample group information and instead identifies DMRs using a population-scale approach, which allows for a neutral DMR calling that makes no assumptions on sample relatedness. 

We show how MethylScore identifies DMRs from both rather simple mutant-vs.-wildtype comparisons and very complex datasets with hundreds of samples and without a clear, expected sample structure. We show that MethylScore-based DMRs allow to detect new and methylation-sequence-specific sample clusters, for example in the Arabidopsis thaliana 1001 Genomes dataset (Kawakatsu et al., Cell 2016) as well as in rice (Stroud et al., eLife 2013). Finally, we showcase how DMRs identified in this dataset can unravel known and yet unknown genotype-epigenotype associations, illustrating how our pipeline can help resolve new genetic determinants of epigenetic variation. 

We designed our pipeline to be accessible to biology researchers without a background in bioinformatics or computer science. First, the MethylScore pipeline is built in Nextflow and executable in a single command. It is fully containerized, which means that users do not need to deal with complicated installation processes, library dependencies, or version control. Second, input requires nothing but standard read alignment files, a reference genome file, and a simple text-based sample information sheet.

In conclusion, we believe that our pipeline provides a substantial improvement to computational WGBS data analysis and interpretation, and that it makes these types of analyses more accessible to the broader plant science community. We therefore believe that Quantitative Plant Biology would be a highly suitable outlet for the publication of our manuscript.

We are looking forward to hearing from you.

Kind regards,

Claude Becker

---

## [Reviewer Report]

*Comments to Author*: In this manuscript a promising new software pipeline is presented - MethylScore - which can be used to identify differentially methylated regions (DMRs) between plant samples. It is pointed out that many existing tools were not designed specifically with plant genomes in mind and rather designed for e.g. mammals that have different patterns and distributions of DNA methylation. Other limitations of existing tools include heavy multiple testing burden (DSS, methylkit), or the preselection of regions to test, which may miss genuine differences. These plant/mammal differences could impact the analysis and identification of DMRs, for example some regions may be “falsely discarded”. By addressing these limitations of existing tools, MethylScore could be a very attractive tool.

There are several areas where some additional information would help potential users evaluate MethylScore in this manuscript. Hopefully this could be easily added and strengthen the manuscript - points 1,3 and 4 are relatively small additions, point 2 may require some additional work if the DMR data is not already on hand.

1. Previous application of the HMM approach in plants?

Was a similar approach, using the HMM, employed to identify DMRs in Hagmann 2015 (https://doi.org/10.1371/journal.pgen.1004920); and likewise in Wibowo 2018? The Hagmann paper is not cited, so it could be clarified if aspects of MethylScore are adapted from that original paper and what is different? No problem in my mind if parts of MethylScore represent a uniform and documented pipeline based on or inspired by prior work, a clear pipeline is extremely useful, but should this prior work be acknowledged/integrated? Or is this quite different?

2. Comparisons to existing tools

Multiple intriguing reanalysis vignettes using published methylation datasets are presented. These analyses show that MethylScore can replicate prior biological findings and also uncover new biological insights, particularly by virtue of not assigning replication groups before DMR calling. This is terrific. However, what I am also particularly interested in is how MethylScore directly compares to the existing tools. This comparison is noticeably missing in the manuscript. It would be very important to know if MethylScore does effectively address the limitations of existing tools, as this is highlighted as a rationale. For example, if there is less multiple testing burden, does it identify more DMRs? How many of the DMRs overlap between tools? How many DMRs are unique and how many are partially different, and are the differences in size, boundaries, significance score etc. This is key information we would want to know before implementing MethylScore. From our preliminary testing, it looks promising that many DMRs overlap e.g. with DSS DMRs and some are unique. Strongly encourage the authors to consider adding some side-by-side comparison data. This would strengthen the manuscript and could also increase uptake of the pipeline! At the very least, comparisons to the originally reported DMRs in the published papers need to be included. For example, it looks like MethylScore finds more DMRs for the arabidopsis experiment using regenerated plants (Wibowo 2018) compared to the original report (but then did the original paper also use the same HMM approach anyway)?

3. Applicability to larger, more highly methylated genomes and the HMM

We note that there were some adaptations to the implementation of the HMM from the original implementation cited (Molaro 2011), because the original was designed for highly methylated genomes. If so, is MethylScore best for small, less methylated plant genomes (like arabidopsis). Many plants have highly methylated genomes eg maize, even rice is highly methylated in a substantial portion of its genome (Fig S3). It is shown nicely that it works for rice, but it would be helpful to clarify which species or range of methylation levels would be suitable.

- It is mentioned that a “parameter sweep” would likely be required to run this on a different species, a little more guidance would be very helpful!

Secondly, using the two-state HMM, the genome is segmented into methylated and unmethylated regions. I am curious about the distribution of methylation levels for these MR regions? The average length is reported - what is the average and min/max methylation levels for the MR regions in rice and arabidopsis (second paragraph of results)? In particular, for CHH, are regions that are low eg ~20% methylated captured as MRs? A histogram would be very informative.

4. Resource requirements

MethylScore is also provided as a nextflow pipeline with a docker image for reproducible computing. During this review we were also able to download and run MethylScore on some arabidopsis WGBS data and also on a small number of inhouse sorghum (medium sized genome) WGBS samples on our server. We also tried running MethylScore on a larger number of sorghum samples (16) and maize samples, but were unable to run these analyses to completion, possibly due to time, memory or other hardware limitations. While we likely would have been able to solve this with more time, it would be helpful to include some guidance as to the resource requirements for different numbers of samples and different sized genomes in the manuscript. How long did it take to process the 1001 genomes data? And is the tool practical for larger genomes; we are genuinely interested?!

Minor

- It would be very challenging for a novice (stated in abstract) or a “researcher with little computational background” (end of intro) to install and set up the pipeline on their own - probably better to emphasise the reproducibility aspect of using nextflow and containers!

- Fig 6 should be “5” (there is no Fig 5?)

---

## [Reviewer Report]

*Comments to Author*: MethylScore is a NextFlow pipeline for the analysis of whole-genome bisulfite sequencing data. MethylScore has been designed to address the complexity of plant methylation and reduce the reliance on methods optimised for mammalian systems. Its main engine is an iterative k-means clustering algorithm. The authors motivate the problem they are addressing clearly and the approach is well-explained. The examples support the claims about the performance and utility of MethylScore. Furthermore, some interesting biological insights are obtained from their approach. I think this will be a useful tool for the community.

Overall, I found this to be a thoughtful and well-written manuscript. It covers an interesting method development that is clearly explained and presented.

I have only a few minor comments for the authors to consider.

In the introduction the authors state that CpG islands “are non-randomly distributed”. Whilst I’m not disputing that they are non-random, I think the authors mean non-uniformly distributed.

It is stated several times that other software packages are built upon assumptions from mammalian data that are violated by plants. It would help the reader to specifically state what these assumptions and which ones are violated in order to understand exactly what the key developments are and also what the parameters are in the model (see question below on parameters). For instance, plants have three methylation contexts – this presumably leads to an expansion of those contexts in MethylScore and associated methylation rates – how is this handled and what parameters are either learnt from the data or need to be set in the parameter file?

The authors make comments about their approach being data-driven, not needing a priori information and thus avoiding inaccurate or erroneous assumptions about the samples. This of course works both ways: using prior knowledge and correct/plausible assumptions often improves the power of the analysis and so neglecting prior information is not necessarily an advantage. My understanding was that including plant-specific prior knowledge, such as the context information, was one of the drivers for this development. It would strengthen the manuscript to give some examples of the type of assumptions they mean and how this affects the results.

On a related note, some comparison with existing methods would greatly strengthen the manuscript. I think this could very nicely show the advantages of their approach and demonstrate the bias introduced by taking parameters from mammalian data. If other existing packages cannot reproduce results that been validated, this would be a powerful demonstration of their software.

The title of Figure 1 (“MethylScore DMR calling clusters samples according to genotype”) took me a while to understand what is meant and I initially thought there was something missing or wrong. It’s actually correct as is but the sentence structure isn’t straightforward as “clusters” is a verb here, whereas “calling” is a noun. Perhaps simplify to avoid confusion.

In the figures, what is the significance of the clustering on the left of the heatmaps? Could this be commented on?

In the PCA plots, are they any features that could be identified that contribute to the principle components? Particularly in Figure 2 where there is such a high eigenvalue for PC1 it would be interesting to learn if any features stand out that contribute most to this eigenvector.

The variance captured by the PCA components in Figure 4 is low, in some cases only around 6% from both PC1 and PC2. This would be good to highlight and given this observation to explain the significance of these plots.

The authors state that there are lots of parameters. From the software description on github I wasn’t clear what all these parameters meant. Some explanation of what are the important parameters and perhaps how this differs from mammalian systems or between species would be helpful. In the Discussion, the authors state that the default parameters work for Arabidopsis and rice but need to be adjusted for optimal results for other species. It would be helpful to explain, exactly which parameters have been tuned and how for Arabidopsis and rice and also how a parameter sweep should be performed. Which parameters will need adjusting? And how is it determined that one has parameters that give “optimal results”? How does this fit with the statement of this being a data-driven approach that doesn’t require prior knowledge?

In the supplemental material, the authors write: “First, to account for the fact that, in contrast to DNA methylation in mammals, plant genomes often only harbor DNA methylation in a small fraction of the genome, the original implementation was adapted to detect hyper-methylated rather than hypo-methylated regions, effectively achieved by inverting the methylation rates.”

I failed to understand exactly what had been done here. It would be useful to define the terms clearly and explain what inverting methylation rates means precisely in this context.

---

## [Reviewer Report]

Dear Olivier,

We would like to submit the revised version of our manuscript entitled “MethylScore, a pipeline for accurate and context-aware identification of differentially methylated regions from population-scale plant WGBS data” (bioRxiv DOI: 10.1101/2022.01.06.475031) for publication in Quantitative Plant Biology.

First, we would like to thank you and the two anonymous reviewers for the constructive comments and suggestions on our manuscript. We believe that we have been able to address all criticisms that were made by the reviewers and that our manuscript has improved considerably in the process. The detailed response is delivered in the attached document. As main improvement to the manuscript, we have included additional analyses that highlight the impact of using a plant-DNA-methylation-specific and context-aware algorithm for differential methylation detection. In our view, this lends credibility to the claim that our tool is advantageous when it comes to analyzing plant WGBS data. In other parts, we have tried to be more accurate and specific in the terminology and phrasing employed. 

We hope that you will find our modifications appropriate and our arguments sufficiently compelling to allow publication of our work in QPB; should you have any further questions, please do not hesitate to contact me. 

We are looking forward to hearing from you.

Kind regards,

Claude

---

## [Reviewer Report]

*Comments to Author*: The authors have thoroughly addressed all the points from the original reviews. I look forwarded to testing and using this new tool!

Minor typo - the paragraph about Supplemental Figure 4 is duplicated at the top of page 7.

---

## [Reviewer Report]

*Comments to Author*: Thank you for your thorough and detailed revision. I believe the authors have addressed all my comments. The manuscript is much improved.

---

## [Reviewer Report]

*Comments to Author*: Dear Claude,

Both reviewers and I are very positive about your tool. The article is now ready to be published.

Congratulations and thank you for choosing QPB,

Olivier